# NinaB and BCO Collaboratively Participate in the β-Carotene Catabolism in Crustaceans: A Case Study on Chinese Mitten Crab *Eriocheir sinensis*

**DOI:** 10.3390/ijms25115592

**Published:** 2024-05-21

**Authors:** Min Zhang, Jingyi Xiong, Zonglin Yang, Boxiang Zhu, Yuting Wu, Xiaowu Chen, Xugan Wu

**Affiliations:** 1Key Laboratory of Exploration and Utilization of Aquatic Genetic Resources, Ministry of Education, Shanghai Ocean University, Shanghai 201306, China; zhangmin199502@163.com (M.Z.); xiongjy0401@163.com (J.X.); zonglinyang701@163.com (Z.Y.); m210100312@st.shou.edu.cn (B.Z.); wyt1632406010414@163.com (Y.W.); 2Shanghai Collaborative Innovation for Aquatic Animal Genetics and Breeding, Shanghai Ocean University, Shanghai 201306, China; 3Centre for Research on Environmental Ecology and Fish Nutrition of the Ministry of Agriculture, Shanghai Ocean University, Shanghai 201306, China

**Keywords:** crustacean, carotenoid metabolism, carotenoid cleavage oxygenases, RNA interference, gene characterization

## Abstract

Carotenoid cleavage oxygenases can cleave carotenoids into a range of biologically important products. Carotenoid isomerooxygenase (NinaB) and β, β-carotene 15, 15′-monooxygenase (BCO1) are two important oxygenases. In order to understand the roles that both oxygenases exert in crustaceans, we first investigated *NinaB-like* (*EsNinaBl*) and *BCO1-like* (*EsBCO1l*) within the genome of Chinese mitten crab (*Eriocheir sinensis*). Their functions were then deciphered through an analysis of their expression patterns, an in vitro β-carotene degradation assay, and RNA interference. The results showed that both *EsNinaBl* and *EsBCO1l* contain an RPE65 domain and exhibit high levels of expression in the hepatopancreas. During the molting stage, *EsNinaBl* exhibited significant upregulation in stage C, whereas *EsBCO1l* showed significantly higher expression levels at stage AB. Moreover, dietary supplementation with β-carotene resulted in a notable increase in the expression of *EsNinaBl* and *EsBCO1l* in the hepatopancreas. Further functional assays showed that the *EsNinaBl* expressed in *E. coli* underwent significant changes in its color, from orange to light; in addition, its β-carotene cleavage was higher than that of *EsBCO1l*. After the knockdown of *EsNinaBl* or *EsBCO1l* in juvenile *E. sinensis*, the expression levels of both genes were significantly decreased in the hepatopancreas, accompanied by a notable increase in the redness (*a**) values. Furthermore, a significant increase in the β-carotene content was observed in the hepatopancreas when *EsNinaBl-*mRNA was suppressed, which suggests that *EsNinaBl* plays an important role in carotenoid cleavage, specifically β-carotene. In conclusion, our findings suggest that *EsNinaBl* and *EsBCO1l* may exhibit functional co-expression and play a crucial role in carotenoid cleavage in crabs.

## 1. Introduction

Most animals cannot biosynthesize carotenoids de novo and must obtain them from their diet. In animals, the carotenoid metabolism exerts several vital physiological roles; it exerts antioxidant and immune-enhancing functions and provides retinol precursors [1]. The major metabolic carotenoid conversions that take place in animals include oxidation, reduction, the translation of double bonds, the oxidative cleavage of double bonds, and the cleavage of epoxy bonds [2,3]. At present, research on the carotenoid metabolism in animals primarily focuses on the study of carotenoid cleavage oxygenases (CCOs) [4,5,6,7].

Carotenoid isomerooxygenase and β, β-carotene 15, 15′-monooxygenase (named BCO1 or BCMO), both members of the CCO family, are capable of cleaving various carotenoids into a variety of biologically significant products; this includes retinal, which occurs naturally in the food chain [8]. In mammals, carotenoid oxygenases and isomerases, which are encoded as two separate proteins, collaborate in the bioconversion pathway of retinols. Carotenoid oxygenase BCO1 oxidatively cleaves carotenoids into retinoids, and the retinoid isomerase RPE65 catalyzes the *trans*-to-*cis* isomerization of the C10–C11 double bond of retinoids [9,10,11,12,13]. Meanwhile, neither inactivation nor afterpotential mutant B (NinaB), a protein found in insect genomes, perform both functions via the catalysis of oxidative cleavage at the C15–C15′ double bond and trans-to-cis isomerization at the C11–C12 double bonds of carotenoid substrates [7]. NinaB is required by *Drosophila* for vision and can metabolize carotenoids into visual chromophores [14]. In addition, Chai et al. (2019) discovered that NinaB is involved in the reproduction of the silkworm *Bombyx mori* and the flour beetle *Tribolium castaneum* [15]. Furthermore, a gene named *Carotenoid Oxygenases* and *Retinal Isomerase* (*BmCORI*) has been identified in *B. mori*; this may play a pivotal role in the β-carotene metabolism [16]. To date, only one study has investigated the function of NinaB in crustaceans, and this was carried out using the ridgetail white prawn *Exopalaemon carinicauda*. This study revealed that two genes, namely carotenoid isomeroxygenase-like 1 and 2, are highly expressed in the eyestalk of *E. carinicauda* [17]. In pathogen challenge experiments on *Vibrio parahaemolyticus* and *Aeromonas hydrophala*, the knockout treatment of *E. carinicauda* was performed using *EcNinaB-X1*; this resulted in significantly lower mortality rates than the control, indicating that *EcNinaB-X1* plays a role in immune defense [18].

BCO1, a cytosolic enzyme, cleaves β-carotene into retinal, thus playing an important role in retinoid synthesis [19]. The mutations that occur in this gene have been verified in some vertebrates, demonstrating their potential to influence the efficiency of carotenoid conversion and their content [20,21]. However, very few reports on the activities of BCO1 in aquatic animals could be found. For instance, *bcox* knockdown in zebrafish resulted in severe malformation and reduced pigmentation [12]. *BCMO1*-mRNA in the nematode *Caenorhabditis elegans* is localized to intestinal cells, which are the tissues used to detect retinal [22]. In mollusks, *βCDOX* was expressed in various tissues of the pearl oyster *Pinctada fucata martensii*, with the highest level of *βCDOX* found in the hepatopancreas [23]. Moreover, although the ability of *PyBCO-like 1* to cleave β, β-carotene in yesso scallop *Patinopecten yessoensis* has been confirmed, the specific mechanism implicated in carotenoid cleavage in the adductor muscle remains unknown [24]. As insect genomes only contain NinaB as a carotenoid oxygenase, there are few studies on the expression and function of BCO in arthropods. However, both *BCO* and *NinaB* genes have been identified in the genome of the ridgetail white prawn *E*. *carinicauda*, and six BCOs were found to be significantly expressed in its stomach and hepatopancreas; this suggests that carotenoid cracking may occur in these two tissues [17].

Although previous studies have highlighted the crucial roles exerted by carotenoid oxygenases and isomerases in the carotenoid metabolism of animals, their functions remain largely unclear in crustaceans. Chinese mitten crab *E. sinensis* is an important economic crustacean; it is particularly popular due to its delicate flavor, red carapace, and deep yellow hepatopancreas [25,26,27]. With regard to the edible tissues of *E. sinensis*, its hepatopancreas and ovaries are rich in carotenoids [28]. However, very limited information about the CCOs present in this species is available. A recent study identified six CCO genes from the *E. sinensis* genome, and their expression pattern and evolution were further investigated [29]. Among these genes, one *EsBCO1-like* gene (*EsBCO1l*) and one *EsNinaB*-*like* gene (*EsNinaBl*) were screened and validated based on bioinformatics. However, the exact functions exerted by NinaB and BCO1 during the carotenoid catabolism of *E. sinensis* remain unclear.

In this study, the expression patterns of *EsNinaBl* and *EsBCO1l* were first investigated, and their recombinant proteins were then obtained from β-carotene-producing *Escherichia coli*; this was followed by biochemical characterization. Furthermore, the knockdown of *EsNinaBl* and *EsBCO1l* was conducted to further elucidate their functions during carotenoid cleavage. Moreover, the effects of dietary β-carotene supplementation on the gene expression levels of the hepatopancreas were further investigated for *EsNinaBl* and *EsBCO1l*. This study explores, for the first time, the differences between these two carotenoid cleavage enzymes, thus filling a gap in the research on carotenoid cleavage in crabs and contributing to revealing the carotenoid metabolism in crustaceans.

## 2. Results

### 2.1. Sequence Characterization and Phylogenetic Analysis of NinaB and BCO1

The multiple sequence alignment analysis revealed that EsNinaB shared a remarkably high degree of similarity with the NinaB obtained from closely related species (Appendix A). The phylogenetic analysis showed that the NinaB of *E. sinensis* formed a branch with *Homarus americanus* and that they constituted a cluster with the NinaB present in other insects (Figure 1). Simultaneously, the analysis highlighted that the BCO1 of *E. sinensis*, together with other crustacean BCO1s, formed a separate cluster that was closely associated with NinaB; thus, a supercluster with a bootstrap support of 76% was formed. Furthermore, the conservation motif analysis unveiled the presence of motifs 1–9 in all vertebrates, while insecta and malacostraca (excluding *Procambarus clarkii*) lacked motif 9. Additionally, compared to the NinaB in insecta and BCO1 in malacostraca, the NinaB in malacostraca lacked motif 7. The domain analysis revealed that a RPE65 domain exists in both BCO1 and NinaB across all species.

### 2.2. Gene Expression Patterns of EsNinaBl and EsBCO1l

The gene expression patterns of *EsNinaBl* and *EsBCO1l* were analyzed in different tissues of *E. sinensis* and at different molting stages. The results showed that *EsNinaBl* was strongly expressed in the hepatopancreas (*p* < 0.05) and that *EsBCO1l* exhibited the highest expression in the epidermal layer; this was followed by the eyestalk and hepatopancreas (*p* < 0.05). Both genes were also expressed in other tissues (Figure 2A). Moreover, the expression of *EsNinaBl* increased during stage C (molting preparation period). Later, *EsNinaBl* was continuously expressed at a low level until the next molting period. As for *EsBCO1l*, its expression was highest during the AB stage (*p* < 0.05); this was followed by a significant decrease during stage C and a noticeable upward trend during stages D and E (Figure 2B).

### 2.3. Recombinant EsBCO1l and EsNinaBl Could Cleave β-Carotene

To further investigate the function of *EsBCO1l* and *EsNinaBl* in cleaving carotenoids, the recombinant *EsBCO1l*/*EsNinaBl*-pGEX-4T plasmid was transformed into *E. coli* strains, which accumulate β-carotene and exhibit a yellow color. In comparison to the *E. coli* controls that were transformed with the vector alone, the bacteria expressing *EsBCO1l* exhibited no significant change in color; meanwhile, evident decolorization was observed in the β-carotene-accumulated bacteria expressing *EsNinaBl* (Figure 3A). Furthermore, we investigated the ability of *EsBCO1l* and *EsNinaBl* to degrade β-carotene. It was evident that *EsNinaBl* exhibited a higher capacity for β-carotene degradation than *EsBCO1l* (Figure 3B).

### 2.4. Knockdown of EsNinaBl and EsBCO1l Increases β-Carotene Deposition in the Hepatopancreas

It was observed that, after 15 days of interference, there was a significant reduction in the expression of *EsNinaBl* upon *EsNinaBl* knockdown (*p* < 0.05). Likewise, a parallel occurrence was observed in the dsEsBCO1-like group (Figure 4B). Moreover, for the crabs injected with the dsRNA of *EsNinaBl* and *EsBCO1l*, the colors of the hepatopancreas changed significantly compared to those of the control group (Figure 4C). Figure 4D shows the color parameters of the hepatopancreas of juvenile crabs after dsEsNinaB-like and dsEsBCO1-like injections. The juveniles administered dsEsNinaB-like injections had significantly higher light (*L**), red (*a**), and yellow (*b**) values than the controls (*p* < 0.05). Meanwhile, the color values of the dsEsBCO1-like group only exhibited a significant difference in their *a** values compared to the control (*p* < 0.05). Notably, the *L** and *b** values of the dsEsNinaB-like group were significantly higher than those of the dsEsBCO1-like group (*p* < 0.05). Considering the apparent change in the color parameters of the hepatopancreas in the dsEsNinaB-like group, we determined their carotenoid content using HPLC analysis. There was a twofold increase in β-carotene in the dsEsNinaB-like group compared to the control and dsEsBCO1-like groups (Figure 4E).

### 2.5. Carotenoid Intake Influences the Expression of NinaBl and BCO1l

As shown in Figure 5, a significant increase in the mRNA expression of *EsNinaBl* and *EsBCO1l* with the increase in dietary β-carotene supplementation was observed. For *EsNinaBl*, the addition of 60 ppm of β-carotene did not result in a significant increase in its expression; meanwhile, there was a remarkable fivefold increase in the expression of *EsBCO1l* (*p* < 0.05). When the β-carotene supplementation reached 120 ppm, a substantial elevation in the expression of *EsNinaBl* was observed, showing a three- to fourfold increase compared to the control and β-car-60 groups. Similarly, the expression of *EsBCO1l* was upregulated significantly, showing values that were approximately 15 times higher than the control and around 2.5 times higher than the β-car-60 group.

## 3. Discussion

Crustaceans accumulate carotenoids through dietary intake, and carotenoid supplementation can provide significant health benefits, such as coloration and disease resistance [30]. Once carotenoids enter the body, they can be broken down into a series of crucial compounds, with carotenoid oxygenase and carotenoid isomerase playing essential roles in this process. BCO1 catalyzes the symmetrical cleavage of the C15–C15′ double bonds of carotenoids that exhibit provitamin A activity, such as β-carotene; it then converts them into colorless retinol [6,10]. NinaB, initially discovered in the fruit fly *Drosophila melanogaster*, catalyzes the trans–cis isomerization of the C11–C12 double bonds and the oxidative cleavage of the C15–C15′ double bonds in the main chain of carotenoids; visual chromophores are thus directly produced [7]. Research on the honeycomb moth *Galleria mellonella* has confirmed that NinaB possesses a dual function, performing both oxidative cleavage and isomerization [31]. It is noteworthy that both NinaB and BCO1 are capable of cleaving the C15–C15′ double bonds in the main chain of carotenoids; however, it remains unclear how they differ in this process. Both genes exhibit a typical RPE65 domain and belong to the CCO family. The evolutionary analysis showed that BCO1s and NinaBs are divided into two clades with high bootstrap support, indicating a relatively stable evolutionary relationship between them. In *E. carinicauda*, NinaB-X1 was also not classified as the same branch of BCO1, similar to the results of this study [18]. Compared to NinaB in insects and BCO1 in crustaceans, NinaB in crustaceans lacks motif 7, suggesting a potential evolutionary association. Furthermore, the structural domain analysis revealed a consistent, singular, and conserved RPE65 domain in both BCO1 and NinaB across all species. Similarly, a previous analysis of the CCO family in *P. trituberculatus*, *E. sinensis*, *Penaeus chinensis*, and *Hyalella azteca* indicated that these CCO genes have highly similar structures [29].

The activity and expression of carotenoid cleavage enzymes were found in several tissues of various vertebrate species [32]. However, research on these enzymes in crustaceans is currently limited. An investigation of the *BCO* genes in *E. carinicauda* revealed that *BCO6* exhibits a high level of expression in the heart, while other *BCO* genes demonstrate elevated levels of expression in the hepatopancreas and stomach [17]. For *EsBCO1l*, we found that it exhibited high levels of expression in the epidermis layer, eyestalk, and hepatopancreas. Liu et al. found that the *CCO3* (i.e., *BCO1*) gene was also highly expressed in the hepatopancreas of *E. sinensis* [29]. Moreover, *EsNinaBl* was highly expressed in the hepatopancreas in this study. These findings suggest that the hepatopancreas may be a crucial tissue in carotenoid digestion and cleavage in crustaceans. Additionally, studies on bivalves such as triangle sail mussel *Hyriopsis ctumingii* and pearl oyster *Pinctada martensii* have shown that *BCO1* is predominantly expressed in the hepatopancreas, where the total carotenoid content is significantly higher than in other tissues [23,33]. This indicates that the hepatopancreas might be a crucial organ in carotenoid metabolism. Interestingly, in our study, *EsBCO1l* and *EsNinaBl* exhibited high expression levels in the eyestalk, which aligns with a previous study that reported the elevated expression of carotenoid isomerooxygenase-like 1 and 2 in the eyestalks of *E. carinicauda* [17]. In *Drosophila melanogaster*, NinaB has been identified as a key component in the production of visual pigments [31]. Therefore, we hypothesize that *BCO1* and *NinaB* may have additional biological functions in other tissues.

Carotenoids play a critical role in the reproduction and growth of crustaceans [34,35]. Molting, a pivotal biological process in crustaceans, necessitates the substantial input of energy for water absorption and the swelling of the exoskeleton, which is intricately linked to both growth and gonad development [36,37]. The hepatopancreas serves as the primary energy source for the molting process, contributing to energy homeostasis and ensuring survival post-molting [36,38,39]. In comparison to other molting stages, the expression level of *EsBCO1l* was significantly higher in the AB stage; this may be due to the fact that the crabs had just completed molting and the new exoskeleton had not fully hardened; more carotenoid metabolites were thus required to promote growth [40]. Unlike *EsBCO1l*, stage C exhibited a substantial increase in the expression of *EsNinaBl* in the hepatopancreas. Although carotenoids are not directly required in stage C, we speculate that crustaceans undergoing exoskeleton shedding and the formation of a new exoskeleton may benefit from the antioxidant and immune-modulating functions of carotenoids [41]. With regard to the pathogen challenge of *E. carinicauda*, the roles of *NinaB-X1* and *BCO2* in immune defense have been confirmed; this is attributed to the accumulation of carotenoids [18,42]. In addition, our bioinformatic analysis revealed that *EsBCO1l* and *EsNinaBl* have RPE65 domains, which exert the oxidoreductase effects of a single donor with the incorporation of molecular oxygen or two atoms of oxygen [43]. Therefore, *EsBCO1l* and *EsNinaBl* in the hepatopancreas may be involved in immune function, helping to alleviate oxidative stress during the molting process via the accumulation of carotenoids. Consequently, both NinaB and BCO play pivotal roles in the vital physiological process of molting in crabs.

In this study, it was found that the hepatopancreas is the main tissue used to store and metabolize carotenoids [35]. Given that carotenoid cleavage oxygenase families (CCOs) are capable of cleaving carotenoids enzymatically [4], it is therefore reasonable to hypothesize that *EsNinaBl* and *EsBCO1l*, which belong to the CCO family, play a role in the carotenoid cleavage process that takes place in the hepatopancreas of crabs. In this study, we constructed the recombinant plasmids pGEX-*EsNinaBl* and pGEX-*EsBCO1l* in vitro. These plasmids were co-expressed by inserting them into *E. coli*, which can produce β-carotene. The noticeable decrease in yellow coloration and the reduction in the β-carotene content indicated that *EsNinaBl* possesses a stronger ability to cleave β-carotene compared to *EsBCO1l*. Similar findings were reported in a study on scallops, where BCO was found to cause a lightening of the color of *E. coli*-producing carotenoids [24].

To further investigate the potential effects of pGEX-*EsNinaBl* and pGEX-*EsBCO1l*, we conducted in vivo RNA interference on juvenile crabs. After 15 days of silencing *NinaBl*, a slight increase in *L** and *b** was observed in the dsNinaB-like group compared to the control. Similarly, the deletion of the *NinaB-X1* and *BCO* genes in *E. carinicauda* resulted in a significant change in the hepatopancreas to a red color [18,42]. Additionally, in comparison to the dsBCO1-like group, the dsNinaB-like group showed significantly higher *L** and *b** values. This suggests that knocking down the *NinaBl* gene has a more pronounced impact on the coloration of the hepatopancreas in crabs compared to *BCO1l*. This finding was consistent with the in vitro results. Moreover, we observed a substantial increase in the β-carotene content in the hepatopancreas tissues of the dsNinaB-like experimental group; these results were approximately double those of the control group and the dsBCO1-like group. Earlier studies reported that knocking down *BCO-like 6* in the hepatopancreas tissues of *E. carinicauda* not only induced a change in coloration but also led to a fivefold increase in the β-carotene content [17]. Therefore, the integration of in vitro and in vivo experiments indicates that NinaB exhibits a more pronounced ability to cleave β-carotene.

Numerous studies have demonstrated that the carotenoid content and color of crustaceans are enhanced significantly when they are fed diets with carotenoid supplementation [30,44,45]. In animals, dietary β-carotene serves as the substrate for the production of vitamin A, which is then converted into the chromophore [46]. The critical step in the conversion of β-carotene to vitamin A is the centric cleavage performed by BCO; in *Drosophila*, this is encoded by the *ninaB* gene [14]. Therefore, we conducted a β-carotene feeding experiment to investigate whether different intakes of dietary β-carotene would affect the expression of *EsNinaBl* and *EsBCO1l* in vivo. The results revealed a significant increase in the expression levels of *EsNinaBl* and *EsBCO1l* with the escalating addition of β-carotene. In particular, there was a notable increase in the expression level of *EsBCO1l* in β-car-120, which was almost 15 times greater than that of the control. We proposed that crabs experience the negative feedback regulation of retinoid homeostasis, in which nuclear receptor transcription factors regulate the gene expressions of carotenoid cleavage oxygenases to control the conversion of carotenoids and the production of retinoids [47]. This regulation may prevent the excessive production of vitamin A when the supply of provitamin A carotenoids is abundant and promote its production in deficient conditions. BCO1, a key enzyme involved in the formation of vitamin A, catalyzes the centric cleavage of β-carotene to yield retinaldehyde [48,49]. To sum up, it is reasonable to speculate that EsNinaB and EsBCO1 are functionally co-expressed; the two gene products may collaborate primarily in crab hepatopancreas to perform carotenoid cleavage.

## 4. Materials and Methods

### 4.1. Animals

Juvenile and adult crabs were sourced from the outer ponds of Chongming Research Station at Shanghai Ocean University, China, and maintained in an indoor facility with a circulating freshwater system. The crabs were fed daily; residual feed and feces were removed nightly. Hepatopancreas tissues were harvested at various molting stages, including AB (postmolt), C (intermolt), D (premolt), and E (ecdysis). Additionally, tissues from the gills, eyestalks, muscles, midgut, hindgut, stomach, and heart were collected from adult crabs and preserved at −80 °C for subsequent comparative expression analyses. All sampling procedures adhered to the ethical guidelines established by the Committee on Experimental Animal Management at Shanghai Ocean University. The approval code is SHOU-DW-2018-033, and the approval date is 26 January 2018.

### 4.2. RNA Extraction and cDNA Synthesis

The total RNA was extracted from the hepatopancreas of the crabs using RNAex Pro Reagent (AGBio, Changsha, China) following the manufacturer’s protocol. The quality of RNA was assessed using NanoDrop 2000 (Thermo Fisher Scientific, Waltham, MA, USA) and 1% agarose gel electrophoresis. Subsequently, cDNA was synthesized from the extracted RNA using the PrimeScript RT Reagent Kit (AGBio, Changsha, China), adhering to the manufacturer’s instructions.

### 4.3. Gene Bioinformatic Analysis

Based on the genome data of *Eriocheir sinensis* (accession no. ASM2467909), we identified the coding sequences (CDS) for NinaB (XM_050837057.1) and BCO1 (XM_050877727.1). We subsequently confirmed the full-length cDNA sequences of *NinaB-like* (*EsNinaBl*) and *BCO1-like* (*EsBCO1l*) in *E. sinensis*. The protein domains were predicted using Interpro (https://www.ebi.ac.uk/interpro/, accessed on 22 February 2024), and a motif analysis was performed using the MEME Suite (https://memesuite.org/meme/doc/meme.html, accessed on 22 February 2024). Multiple sequence alignments were carried out using MEGA11.0, and phylogenetic trees were constructed using the Maximum Likelihood method, with the visualization provided by TBtools-II software version 2.069.

### 4.4. In vivo Verification of EsNinaBl and EsBCO1l Functions

Gene fragments of *EsNinaBl*/*EsBCO1l* were inserted into the L4440 vector to construct interference vectors. These recombinant plasmids were transformed into HT115 competent cells (MKBio, Shanghai, China) and induced with IPTG to produce dsRNA that would target the specific fragments. The dsRNA was purified and diluted to a concentration of 1 μg/μL using nuclease-free 1 × PBS.

Juvenile crabs weighing approximately 10 g and in molting stage C were subjected to the dsRNA-mediated gene silencing of *EsNinaBl* and *EsBCO1l*. As outlined in Section 4.1, the crabs were maintained in an indoor container with a circulating freshwater system and were fed daily. Each group contained three replicates, with 10 crabs in each. Each crab received 1 μL of dsRNA or 1 × PBS (as a negative control) per gram of body weight; this was injected into the base of the third walking leg every three days over a period of fifteen days. At the end of the experiment, the survival rates were 60% for the dsRNA group and 80% for the PBS group. Subsequently, hepatopancreas tissues were harvested in order to analyze the expression of *EsNinaBl* and *EsBCO1l* and to detect carotenoids.

### 4.5. In Vitro Verification of EsNinaBl and EsBCO1l Function in E. coli

The pAC-BETAipi plasmid (Miaolingbio, Wuhan, China), which contains β-carotene synthesis genes, was transformed into *E. coli* BL(DE3) cells (Tiangen, Beijing, China). The open reading frames (ORFs) of *EsNinaBl* and *EsBCO1l* were amplified and cloned into the pGEX-4T vector (Miaolingbio, Wuhan, China) using T4 DNA ligase (Sangon, Shanghai, China). These recombinant plasmids were then introduced into *E. coli* strains harboring the pAC-BETAipi plasmid. The colonies that co-expressed β-carotene synthesis genes along with *EsBCO1l*/*EsNinaBl* were induced with 1 mM of IPTG. The phenotypic changes in the *E. coli* colonies were assessed, and those with only the pAC-BETAipi plasmid, those with both pAC-BETAipi and the recombinant pGEX-4T-*EsBCO1l*/*EsNinaBl* plasmids, and those with the pGEX-4T vector alone were compared.

### 4.6. Carotenoid Feeding

Healthy crabs, averaging 100 g in weight, were sourced from Chongming Research Station, Shanghai Ocean University, China. The crabs were individually cultured in boxes with shelters within a freshwater recirculating system and acclimated for one week using a carotenoid-free commercial crab feed.

In the formal experiment, 21 crabs were cultured in controlled tanks at 24 °C and divided into three groups of seven: one control and two β-carotene diet groups. β-carotene, sourced from Zhejiang NHU Co., Ltd. (Shaoxing, China), was administered at concentrations of 60 ppm and 120 ppm by incorporating it into the diet. The control group received a base diet devoid of any carotenoids. The details of the diet are provided in Table 1. The crabs were fed between 1.5 and 3.0% of their body weight daily at 18:00, with waste and uneaten feed removed by the following morning. The 28-day feeding trial concluded with the random selection of five crabs from each group for the collection of hepatopancreas tissue, which was immediately frozen in liquid nitrogen and stored at −80 °C for analysis.

### 4.7. Quantitative PCR

The gene expression levels were quantified using CFX384^TM^ Real-Time System (Bio-Rad, Hercules, CA, USA). The reactions utilized 1 μL of cDNA with the SYBR Green Premix Pro Taq HS qPCR Kit (AGBio, Wuhan, China) in a 10 μL total reaction volume; this comprised 5 μL of 2 × SYBR Green Pro Taq HS Premix, 0.2 μL of each primer, and 3.6 μL of nuclease-free water. All qRT-PCR analyses were conducted with five biological replicates, and each sample was run in triplicate. β-actin served as the reference gene for normalization. The primer sequences that were used are listed in Table 2. The relative gene expression was calculated using the 2^−ΔΔCt^ method [50].

### 4.8. Carotenoid Extraction and HPLC Analysis

The total carotenoids were extracted from tissues using acetone and then filtered. The optical density (OD) of the extracts was measured using a UV–Visible spectrophotometer (T6 New Century, Beijing Purkinje General Instrument Co., Ltd., Beijing, China) to estimate the β-carotene levels. The β-carotene content was quantified using an Agilent 1260 high-performance liquid chromatography (HPLC) system (Agilent Technologies Inc., Santa Clara, CA, USA) following the methodology described by Peng et al. [51]. The Agilent 1260 HPLC system utilized a YMC™ Carotenoid C30 column (4.6 × 150 mm, 3 μm particle size, YMC Co., Ltd., Kyoto, Japan). The gradient mobile phases consisted of phase A (methyl alcohol/methyl tert-butyl ether/formic acid at a ratio of 3:2:0.01, *v*/*v*/*v*) and phase B (alcohol/triethylamine at a ratio of 100:0.04, *v*/*v*). Details of the procedures used to perform the carotenoid analysis in crab tissues can be found in the methodology described by Long et al. [28].

### 4.9. Statistical Analysis

Statistical analyses were performed using one-way analysis of variance (ANOVA), followed by Tukey’s test, with SPSS 22.0 software. The results were deemed statistically significant at *p*-values less than 0.05. Values are presented as the means ± standard error (SE). All graphs were generated using GraphPad Prism version 7.03.

## 5. Conclusions

To the best of our knowledge, this study is the first to explore the functions of *NinaB* and *BCO1* during carotenoid cleavage using comprehensive approaches. Both *EsNinaBl* and *EsBCO1l* are highly expressed in the hepatopancreas, but they have different expression patterns during the molting cycle of *E. sinensis*. The assessment of the catalytic activity of the recombinant protein in *E. coli* showed that *EsNinaBl* had a stronger catalytic efficiency than *EsBCO1l* for carotenoid cleavage. The knockdown of *EsNinaBl* or *EsBCO1l* led to an increase in β-carotene deposition in the hepatopancreas of juvenile *E. sinensis*, and *EsNinaBl* exhibited a significantly higher ability to degrade β-carotene than *EsBCO1l*. Moreover, β-carotene feeding led to a significant increase in the expression of both genes in the hepatopancreas. Taken together, these findings suggest that *NinaB* and *BCO1* collaborate during carotenoid cleavage in *E. sinensis*. The present results offer initial insights into the importance of *NinaB* and *BCO* in carotenoid cleavage in crabs and provide valuable information regarding the regulation of the carotenoid metabolism in crustaceans.

## Figures and Tables

**Figure 1 ijms-25-05592-f001:**
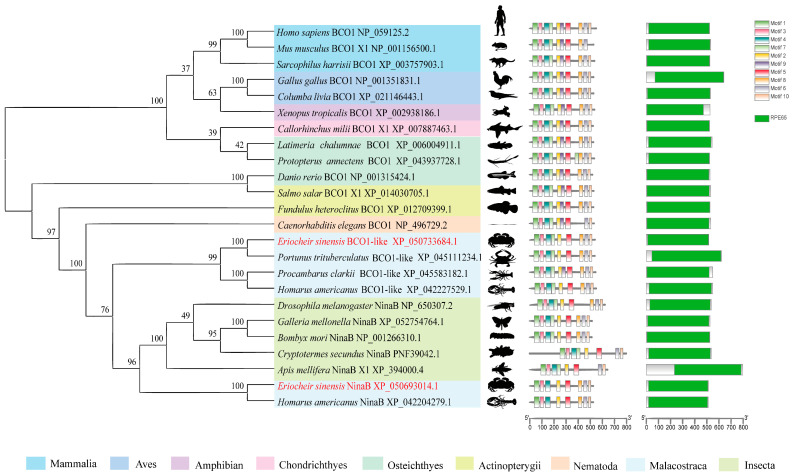
Alignment of the phylogenetic tree of NinaB and BCO1. Values on the line of the phylogenetic tree are bootstrap values showing the percentage confidence in relatedness.

**Figure 2 ijms-25-05592-f002:**
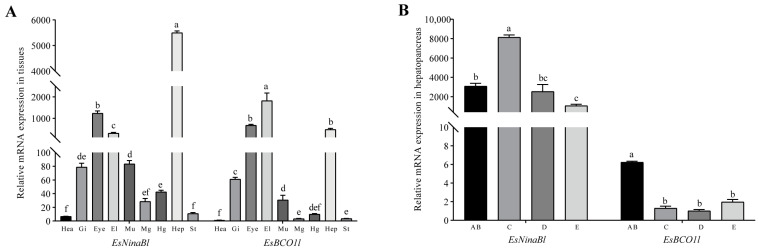
Gene expression patterns of *EsNinaBl* and *EsBCO1l* in *E. sinensis.* (**A**) Gene expression levels of *EsNinaBl* and *EsBCO1l* in the different tissues of juvenile crabs. Hea: heart; Gi: gill; Eye: eyestalks; El: epidermal layer; Mu: muscle; Mg: midgut; Hg: hindgut; Hep: hepatopancreas; St: stomach. (**B**) The gene expression pattern of *EsNinaBl* and *EsBCO1l* in the hepatopancreas during the molting cycle of juvenile crabs. AB: postmolt; C: intermolt; D: premolt; E: ecdysis process. The amount of *EsNinaBl* and *EsBCO1l* mRNA was normalized to the β-actin transcript level. The data are shown as the means ± SEM (*n* = 5). The columns with different letters on the top indicate that they are significantly different for the same parameter (*p* < 0.05).

**Figure 3 ijms-25-05592-f003:**
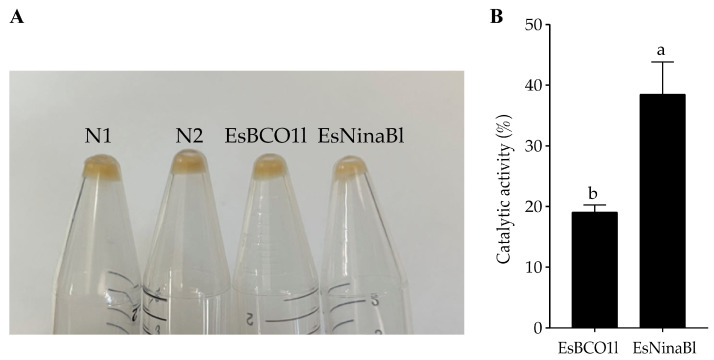
Induced expression of pGEX-NinaBl and pGEX-BCO1l plasmids in *E. coli* strains engineered to accumulate β-carotene. (**A**) N1: pellet from the pGEX-4T vector in *E. coli* without expression; N2: pellet from the pGEX-4T vector with expressed *E. coli*; EsBCO1l: pellet from β-carotene-producing and pGEX-4T-*EsBCO1l*-expressed *E. coli*; EsNinaBl: pellet from β-carotene-producing and pGEX-4T-*EsNinaBl-*expressed *E. coli*. (**B**) Comparison of the degradation capacity of β-carotene among EsBCO1l and EsNinaBl. The columns with different letters on the top indicate that they are significantly different for the same parameter (*p* < 0.05).

**Figure 4 ijms-25-05592-f004:**
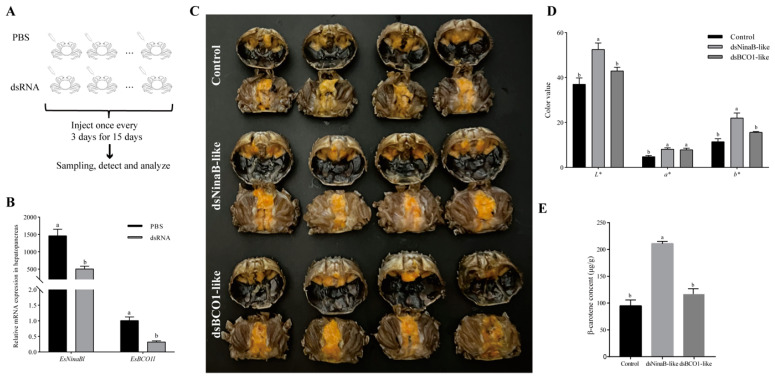
Effects of dsRNA injection on gene expression (**B**), appearance characteristics (**C**), color parameters (**D**), and β-carotene contents (**E**) in the hepatopancreas of *E. sinensis*. (**A**) The diagram of this RNA interference experiment. The data are shown as the means ± SEM (*n* = 5). The columns with different letters on the top indicate that they are significantly different for the same parameter (*p* < 0.05).

**Figure 5 ijms-25-05592-f005:**
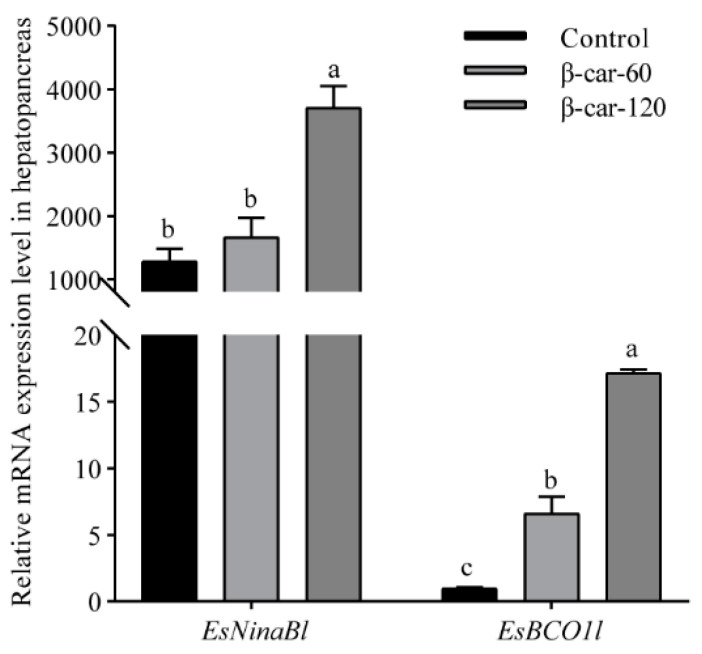
Effects of dietary β-carotene supplementation on gene expression levels of *EsNinaBl* and *EsBCO1l* in the hepatopancreas of *E. sinensis*. The amount of *EsNinaBl* and *EsBCO1l* mRNA was normalized to the β-actin transcript level. The data are shown as the means ± SEM (*n* = 5). The columns with different letters on the top indicate that they are significantly different for the same parameter (*p* < 0.05).

**Table 1 ijms-25-05592-t001:** The formulation, proximate composition, and actual carotenoid concentration of experimental diets.

Items	Control	β-Car-60	β-Car-120
Ingredients/%			
Soybean meal	19.90	19.90	19.90
Rapeseed meal	8.00	8.00	8.00
Peanut meal	8.00	8.00	8.00
Fish meal	22.00	22.00	22.00
Yeast extract	2.00	2.00	2.00
High protein flour	16.55	16.55	16.55
Fish slurry	8.00	8.00	8.00
Soy lecithin	2.00	2.00	2.00
Fish oil	4.00	4.00	4.00
Rapeseed oil	1.50	1.50	1.50
Soybean oil	1.50	1.50	1.50
Carboxymethylcellulose	3.00	3.00	3.00
Cholesterol	0.40	0.40	0.40
Choline chloride	0.40	0.40	0.40
Vitamin premix ^1^	0.20	0.20	0.20
Vitamin C palmitate	0.15	0.15	0.15
Vitamin E	0.05	0.05	0.05
Mineral premix ^2^	0.25	0.25	0.25
Ca(H_2_PO_4_)_2_	1.20	1.20	1.20
Taurine	0.30	0.30	0.30
Cellulose	0.60	0.544	0.488
Synthetic β-carotene	0.00	0.056	0.112
Proximate composition/%			
Moisture	8.76 ± 0.05	8.93 ± 0.06	8.83 ± 0.02
Crude protein	40.51 ± 0.05	40.30 ± 0.21	40.17 ± 1.64
Crude lipid	12.05 ± 0.31	11.98 ± 0.11	12.11 ± 0.09
Ash	9.44 ± 0.04	9.13 ± 0.03	9.16 ± 0.02
Carotenoid concentration/(mg/kg)			
Total carotenoid	9.30 ± 0.62	67.13 ± 2.41	131.15 ± 1.35
Astaxanthin	0.12 ± 0.03	0.24 ± 0.08	0.23 ± 0.05
β-carotene	3.17 ± 0.64	63.24 ± 1.56	127.33 ± 2.67

Note: ^1^ Vitamin mixture (mg·kg^−1^ diet): retinol acetate 125, cholecalciferol 30, alpha-tocopherol 1300, menadione 35.4, thiamine 100, riboflavin 150, vitamin B_6_ 150, vitamin B_12_ 0.2, ascorbic acid 1050, biotin 4, D-calcium pantothenate 250, folic acid 25, and nicotinamide 300. ^2^ Mineral mixture (mg·kg^−1^ diet): Ca(H_2_PO_4_)_2_ 10,000, KH_2_PO_4_ 4200, NaH_2_PO_4_ 500, FeSO_4_·H_2_O 200, CuSO_4_·5H_2_O 96, ZnSO_4_·H_2_O 360, MnSO_4_·H_2_O 120, MgSO_4_·H_2_O 240, KI 5.4, CoCl_2_·6H_2_O 2.1, and Na_2_SeO_3_ 3.

**Table 2 ijms-25-05592-t002:** Information on the primers used in this study.

Primers	Sequence (5′-3′)	Purpose
*NinaB*-PF	ccggaattcATGTCCACGGACGAGGGTGGA	recombinant protein construction
*NinaB*-PR	ccgggaattcCTGGGCCTCGGCGGGGATGAA	recombinant protein construction
*BCO1-*PF	ccggaattcATGGAGCAGCAACAAGAAGAG	recombinant protein construction
*BCO1*-PR	ccgggaattcGTAAGCGTGGACGTCCTGGCG	recombinant protein construction
*NinaB*-qF	GGATTGACACCTACGACTACTC	qRT-PCR
*NinaB*-qR	CGGACCGTTGTAAATGAGTTGT	qRT-PCR
*BCO1*-qF	CAGCAACAAGAAGAGAACCG	qRT-PCR
*BCO1*-qR	GCGAAGGAATCTGGAACGA	qRT-PCR
β-actin-F	GCATCCACGAGACCACTTACA	qRT-PCR
β-actin-R	CTCCTGCTTGCTGATCCACATC	qRT-PCR
ds*NinaB*-F	tccccgcggCCACGGACGAGGGTGGAAGG	RNA interference
ds*NinaB*-R	ggactagtCCGTCACCCCGACGCCCTGGC	RNA interference
ds*BCO1*-F	tccccgcggGAGCAGCAACAAGAAGAGAAC	RNA interference
ds*BCO1*-R	ggactagtGCAGCGAGAAGGAACAATCA	RNA interference

Note: Lowercase letters in the primers are restriction sites and their protected bases.

## Data Availability

Data are contained within the article and Appendix A.

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
