# Peer review of "NinaB and BCO Collaboratively Participate in the β-Carotene Catabolism in Crustaceans: A Case Study on Chinese Mitten Crab Eriocheir sinensis"

_ijms, 2024, doi:10.3390/ijms25115592_

Round 1

Reviewer 1 Report

Comments and Suggestions for Authors

In this study, authors cloned NinaB-like and BCO1-like from Chinese mitten crab and performed some functional studies. I have read this article, in my opinion, this article is well organized and well written, but I don't think it reflects enough scientific value as it stands.

My concerns are as follows:

1.     The authors should enhance the scientific significance of this study in the manuscript; the current manuscript does not see that this study has significant scientific value.

2.     The methods section of the dsRNA should be more detailed, such as how many crabs were injected, were there any crab deaths in the process.

3.     Figure 1, the phylogenetic tree only traced to crustaceans, do these two genes not exist in lower animals? If so (the author mentioned fruit fly), the phylogenetic tree is incomplete.

4.     The authors should include a brief discussion of the evolution of these two genes, including motifs in the discussion section.

5.     Why didn't the authors express and then purify to obtain these two proteins and perform functional experiments afterwards? Directly in E. coli seems unconvincing.

6.     Figure 4B is too small.

7.     Statistical differences for all other plots are indicated by different letters, but Figure 4D indicated by *.

8.     Please improve Table 1, the current version looks weird.

Author Response

Thank you very much for taking the time to review this manuscript. Please find the detailed responses in the  attachment and the corresponding revisions/corrections highlighted/in track changes in the re-submitted files.

Comments 1: The authors should enhance the scientific significance of this study in the manuscript; the current manuscript does not see that this study has significant scientific value.

Response 1: Thank you for your suggestion. I have modified in line 97-100 as follows: “This study provides the first exploration into the differences of these two carotenoid cleavage enzymes, filling a gap in the research on carotenoid cleavage in crabs and contributing to reveal the carotenoid metabolism in crustaceans”. This statement highlights the importance of these two genes for carotenoid cleavage in crabs, and further enhances the significance of the carotenoid metabolism in crustaceans.

Comments 2: The methods section of the dsRNA should be more detailed, such as how many crabs were injected, were there any crab deaths in the process.

Response 2: Thank you for your suggestion. I have added a detailed description of dsRNA method in line 336 and line 338-340. The number of crabs used in each group is described in line 336, and the survival rates of crabs after the experiment is described in lines 338-340.

Comments 3: Figure 1, the phylogenetic tree only traced to crustaceans, do these two genes not exist in lower animals? If so (the author mentioned fruit fly), the phylogenetic tree is incomplete.

Response 3: Figure 1 illustrates the inclusion of several species with lower taxonomic ranks compared to crustaceans, such as Drosophila melanogaster, Caenorhabditis elegans, and Cryptotermes secundus. These are highlighted in light pink (Nematoda) and light green (Insecta). The phylogenetic tree presented also encompasses a diverse range of taxa, including Mammalia, Aves, Amphibia, Chondrichthyes, Osteichthyes, Actinopterygii, Nematoda, Malacostraca, and Insecta. This diversity makes the tree relatively comprehensive in scope.

Comments 4: The authors should include a brief discussion of the evolution of these two genes, including motifs in the discussion section.

Response 4: Thank you for your suggestion. I have supplemented the evolutionary relationship between NinaB and BCO1 of E. carinicauda, as well as structural analysis of CCOs in four species (P. trituberculatus, E. sinensis, Penaeus chinensis, and Hyalella azteca) to confirm the reliability of our results in line 206-212.

 Comments 5: Why didn't the authors express and then purify to obtain these two proteins and perform functional experiments afterwards? Directly in E. coli seems unconvincing.

Response 5: The color change in E. coli, capable of producing carotenoids, was utilized to assess the carotenoid-cleaving ability of proteins, a technique commonly used to determine carotenoid functionality (Thomas et al., 2020; Li et al., 2019; Chai et al., 2019; von Lintig, 2000, 2001). Several studies referenced in this work support our methodology. This research primarily offers a preliminary comparison of the effectiveness of two proteins in cleaving beta-carotene. Currently, these proteins have been purified, and further functional validation studies are underway. The relevant references are listed below:

  • Thomas, L. D., Bandara, S., Parmar, V. M., Srinivasagan, R., Khadka, N., Golczak, M., Kiser, P. D., von Lintig, J., The human mitochondrial enzyme BCO2 exhibits catalytic activity toward carotenoids and apocarotenoids. Journal of Biological Chemistry 2020, 295, (46), 15553-155565.
  • Li, X.; Wang, S. Y.; Xun, X. G.; Zhang, M. R.; Wang, S.; Li, H. D.; Zhao, L.; Fu, Q.; Wang, H. Z.; Li, T. T.; Lian, S. S.; Xing, Q.; Li, X.; Wu, W.; Zhang, L. L.; Hu, X. L.; Bao, Z. M., A carotenoid oxygenase is responsible for muscle coloration in scallop. Biochimica Et Biophysica Acta-Molecular and Cell Biology of Lipids 2019, 1864, (7), 966-975.
  • Chai, C. L.; Xu, X.; Sun, W. Z.; Zhang, F.; Ye, C.; Ding, G. S.; Li, J. T.; Zhong, G. X.; Xiao, W.; Liu, B. B.; von Lintig, J.; Lu, C., Characterization of the novel role of NinaB orthologs from Bombyx mori and Tribolium castaneum. Insect Biochemistry and Molecular Biology 2019, 109, 106-115.
  • von Lintig, J., Dreher, A.; Kiefer, C.; Wernet, M. F.; Vogt, K., Analysis of the blind Drosophila mutant ninaB identifies the gene encoding the key enzyme for vitamin A formation in vivo. Proceedings of the National Academy of Sciences of the United States of America 2001, 98, (3), 1130-1135.
  • Kiefer, C. , Hessel, S. , Lampert, J. M. , Vogt, K. , Lintig, J. V., Identification and characterization of a mammalian enzyme catalyzing the asymmetric oxidative cleavage of provitamin a. Journal of Biological Chemistry 2001, 276(17), 14110-14116.
  • von Lintig, J., Vogt, K., Filling the gap in vitamin A research: Molecular identification of an enzyme cleaving β-carotene to retinal. Journal of Biological Chemistry 2000, 275, (16), 11915-11920.

Comments 6: Figure 4B is too small.

Response 6: I agree with this comment, and revisions have been made in Figure 4. The original Figure 4B has been enlarged and is now presented as Figure 4C.

 Comments 7: Statistical differences for all other plots are indicated by different letters, but Figure 4D indicated by *.

Response 7: Agreed. I've updated the statistical differences in Figure 4D, now indicated by different letters, and the original Figure 4D has been reassigned as Figure 4E.

Comments 8: Please improve Table 1, the current version looks weird.

Response 8: Thank you for pointing this out. It appears that the format of Table 1 may have been altered during the submission process. I have now revised its format in the manuscript.

Reviewer 2 Report

Comments and Suggestions for Authors

Dear authors,

The article is interesting, based on a large volume of work and is significant for researchers in the field. However, the article presents a series of shortcomings and ambiguities that should be corrected before publication.

Here are my observations.

Abstract section, line 27 - explain the meaning of "a* values". Even if it is an abbreviation, its meaning must be clarified from the very beginning.

Introduction section, line 88 "among them genes" sounds incorrect. Clarify the meaning!

Introduction section, lines 92-93 - it seems that the sentence is incomplete! Please correct.

Results section, Figure 1a - lines 112 -116. The amino acid alignment should be presented more synthetically in the content of the article or should be moved as Supplementary Material sooner. I think that for the interested reader, Figure 1 b is sufficient, with one condition that it be made readable, because the font is too small and almost impossible to understand. Please modify!

Discussion section, lines 240-245 - please explain this statement more clearly and with more convincing arguments.

Line 284 - what does "uclear" mean?

Discussion section, lines 240-245 - please explain this statement more clearly and with more convincing arguments.

Line 284 - what does "uclear" mean?

Lines 222-225 - the phrase is confused. Please clarify the meaning.

Conclusions section. I think it would be useful to address the practical importance of the activity of these genes, especially in the Discussions. Also, in the Conclusions, it is necessary to emphasize the original contribution to the knowledge of the activity of these genes.

With best regards!

Comments on the Quality of English Language

Only minor issues like typographical errors.

Author Response

Comments 1: Abstract section, line 27 - explain the meaning of "a* values". Even if it is an abbreviation, its meaning must be clarified from the very beginning.

Response 1: Thank you for pointing this out. I agree with this comment. Therefore, I have added the meaning of "a*" in line 27.

Comments 2: Introduction section, line 88 "among them genes" sounds incorrect. Clarify the meaning!

Response 2: Agree. I have, accordingly, revised it to "among these genes" to refer to the previously mentioned "6 CCO genes" in line 88.

Comments 3: Introduction section, lines 92-93 - it seems that the sentence is incomplete! Please correct.

Response 3: Thank you for pointing this out. I have revised this sentence in line 92-94.

Comments 4: Results section, Figure 1a - lines 112 -116. The amino acid alignment should be presented more synthetically in the content of the article or should be moved as Supplementary Material sooner. I think that for the interested reader, Figure 1 b is sufficient, with one condition that it be made readable, because the font is too small and almost impossible to understand. Please modify!

Response 4: Thank you for your suggestion. Figure 1A has been moved as Supplementary Material and renamed as Figure S1, and is temporarily displayed in line 554. The original Figure 1B has been enlarged and is now presented as Figure 1.

Comments 5: Discussion section, lines 240-245 - please explain this statement more clearly and with more convincing arguments.

Response 5: Thank you for pointing this out. I have supplemented the findings regarding NinaB-X1-KO and BCO2-KO in pathogen resistance through carotenoid accumulation, confirming their roles in immune defense, as indicated in lines 244-251.

Comments 6: Line 284 - what does "uclear" mean?

Response 6: The "uclear" should be changed to "nuclear," which was inadvertently deleted from the manuscript during a previous revision. This error has been corrected in line 291.

Comments 7: Lines 222-225 - the phrase is confused. Please clarify the meaning.

Response 7: Thank you for pointing this out. I have adjusted and modified the phrase accordingly in line 218-220 and line 226-227. The high expression of CCO3 gene in the hepatopancreas elucidates the importance of the hepatopancreas in carotenoid metabolism. Subsequently, the high expression of EsBCO1l and EsNinaBl in the eyestalk suggests that these two genes may have other physiological functions in other tissues.

Comments 8: Conclusions section. I think it would be useful to address the practical importance of the activity of these genes, especially in the Discussions. Also, in the Conclusions, it is necessary to emphasize the original contribution to the knowledge of the activity of these genes.

Response 8: Thank you for your suggestion. I have revised the Conclusions section by adding content regarding the theoretical research and nutritional regulation contributions of these two genes, as indicated in lines 413-415.

Round 2

Reviewer 1 Report

Comments and Suggestions for Authors

I am satisfied with the author's revisions, although I still don't quite understand the scientific value of this paper due to the differences in the research.  However, the author's revision provides good integrity to the article and I think this paper can be accepted at this time.

Author Response

       Thank you for taking the time to review the revised manuscript and for providing your feedback. We appreciate your thorough evaluation and constructive comments.

  We are pleased to hear that you find the author's revisions satisfactory and that they have contributed to the integrity of the manuscript. We understand your point regarding the perceived scientific value of the manuscript, considering the differences in the research. We acknowledge that while the paper may not align perfectly with certain expectations, we believe it still offers valuable insights and contributes to the existing literature in its own right.

    Based on your assessment and your acknowledgment of the improvements made through revision, we are grateful for your recommendation to accept the paper at this time.

     Once again, we appreciate your dedication to the peer review process and your valuable input in enhancing the quality of the manuscript.